# Characterization and Phylodynamics of Reassortant H12Nx Viruses in Northern Eurasia

**DOI:** 10.3390/microorganisms7120643

**Published:** 2019-12-03

**Authors:** Kirill Sharshov, Junki Mine, Ivan Sobolev, Olga Kurskaya, Nikita Dubovitskiy, Marsel Kabilov, Tatiana Alikina, Momoko Nakayama, Ryota Tsunekuni, Anastasiya Derko, Elena Prokopyeva, Alexander Alekseev, Michael Shchelkanov, Alexey Druzyaka, Alimurad Gadzhiev, Yuko Uchida, Alexander Shestopalov, Takehiko Saito

**Affiliations:** 1Department of Experimental Modeling and Pathogenesis of Infectious Diseases, Federal Research Center of Fundamental and Translational Medicine, 630117 Novosibirsk, Russia; sobolev_i@hotmail.com (I.S.); kurskaya_og@mail.ru (O.K.); nikundefeat@gmail.com (N.D.); ellap@bk.ru (E.P.); al-alexok@ngs.ru (A.A.); shestopalov2@ngs.ru (A.S.); 2Division of Transboundary Animal Disease, National Institute of Animal Health, Tsukuba, Ibaraki 305-0856, Japan; minejun84032@affrc.go.jp (J.M.); nakayamam769@affrc.go.jp (M.N.); tune@affrc.go.jp (R.T.); uchiyu@affrc.go.jp (Y.U.); taksaito@affrc.go.jp (T.S.); 3Genomics Core Facility, Institute of Chemical Biology and Fundamental Medicine, 630090 Novosibirsk, Russia; kabilov@niboch.nsc.ru (M.K.); alikina@niboch.nsc.ru (T.A.); 4School of Biomedicine, Far Eastern Federal University, 690091 Vladivostok, Russia; adorob@mail.ru; 5Laboratory of Virology, Federal Scientific Center of East Asia Terrestrial Biodiversity, 690022 Vladivostok, Russia; 6Laboratory of marine microbiota, National Scientific Center o Marine Biology, 690041 Vladivostok, Russia; 7Laboratory of behavioral ecology, Institute of Animal Systematics and Ecology, 630091 Novosibirsk, Russia; decartez@gmail.com; 8Department of Ecology, Dagestan State University, 367000 Makhachkala, Russia; ali-eco@mail.ru

**Keywords:** avian influenza, rare subtype, H12Nx, multiple reassortant, wild birds, Northern Eurasia, American lineage

## Abstract

Wild waterfowl birds are known to be the main reservoir for a variety of avian influenza viruses of different subtypes. Some subtypes, such as H2Nx, H8Nx, H12Nx, and H14Nx, occur relatively rarely in nature. During 10-year long-term surveillance, we isolated five rare H12N5 and one H12N2 viruses in three different distinct geographic regions of Northern Eurasia and studied their characteristics. H12N2 from the Far East region was a double reassortant containing hemagglutinin (HA), non-structural (NS) and nucleoprotein (NP) segments of the American lineage and others from the classical Eurasian avian-like lineage. H12N5 viruses contain Eurasian lineage segments. We suggest a phylogeographical scheme for reassortment events associated with geographical groups of aquatic birds and their migration flyways. The H12N2 virus is of particular interest as this subtype has been found in common teal in the Russian Far East region, and it has a strong relation to North American avian influenza virus lineages, clearly showing that viral exchange of segments between the two continents does occur. Our results emphasize the importance of Avian Influenza Virus (AIV) surveillance in Northern Eurasia for the annual screening of virus characteristics, including the genetic constellation of rare virus subtypes, to understand the evolutionary ecology of AIV.

## 1. Introduction

Influenza viruses belong to the family of *Orthomyxoviridae* and are divided into subtypes according to their different surface glycoproteins. The 16 main hemagglutinin (HA) subtypes and nine neuraminidase (NA) subtypes have been isolated from a wide range of wild and domestic bird species. Additionally, subtypes H17N10 and H18N11 of influenza-like viruses have been detected in little yellow-shouldered bats [1]. There has been a significant spread of some subtypes—such as H1N1, H3N8, and H4N6—among wild waterfowl, which are considered the main reservoir of low-pathogenic avian influenza (LPAI) viruses. Some subtypes, such as H13Nx and H16N3, are primarily associated with seagulls. Other HA subtype viruses (H2, H8, H12, and H14) occur relatively rarely in nature [2]. However, there is a complex pattern of combinations of internal virus genes belonging to various lines and causing different properties of virus proteins. The process of genome reassortment plays an important role in this. Some subtypes may evolve and acquire a pathogenic phenotype that causes serious disease and is associated with epizootics and pandemics (such as H5Nx and H7Nx) [3,4]. Highly pathogenic avian influenza viruses (HPAIVs) originate from low-pathogenic precursors with the HA subtypes H5 and H7 [3,4]. However, the avian influenza virus subtypes H2, H4, H8, and H14 have been shown to support a highly pathogenic phenotype after the genetically engineered introduction of the polybasic cleavage site into the HA [5]. Therefore, the monitoring and investigation of the pathogenic potential of new and rare subtypes is important for seeking and evaluating the pathogenic potential of new virus variants. Moreover, the evolutionary dynamics of rare viruses vary among subtypes, suggesting different drivers of maintenance in the avian reservoir [6,7]. Although influenza A viruses mostly circulate in wild waterfowl, those that can overcome the interspecies barrier and infect mammals represent the greatest risk of zoonotic spread to humans, playing an important role in the generation of panzootic and/or pandemic viruses.

During an annual AIV surveillance program, we analyzed 1652 samples from wild birds migrating by different flyways (the East Africa–West Asia Flyway, Central Asia Flyway, and East Asia–Australian Flyway) in different parts of Russia during 2017–2018 and isolated 69 viruses, including six H12Nx subtypes, that represented new data for these regions [8]. Transcontinental migrations of numerous wild bird species occur from North Asia and Europe to the Mediterranean, Africa, and Southwest Asia, and also cross from the Baltic and Caspian Seas to the Black and Mediterranean Seas, and from Western Siberia and Kazakhstan to Western Europe and North Africa. Historically, the Azov–Black Sea region is an area of nesting, flight, migratory stops, and wintering for many bird species [9].

Since the H12 subtype is rarely detected in wild birds, in this study, we examined the biological characteristics and complete genome sequence analysis of five H12N5 and one H12N2 viruses and conducted a comparison with the available data in terms of the evolutionary ecology of avian influenza viruses in Northern Eurasia. The data and results presented here further our knowledge of the ecology of rare AIV subtypes. Additionally, we performed a pathogenic risk assessment of these variants regarding their potential threat to humans [10,11].

## 2. Materials and Methods 

### 2.1. Sampling, Virus Isolation, and Cells

Influenza A/H12 viruses were isolated from cloacal swabs collected from wild migratory birds using 10-day-old embryonated chicken eggs, according to standard protocols (World Organisation for Animal Health /WHO), in biosafety level 3 facilities in the Federal Research Center of Fundamental and Translational Medicine (CFTM). Sampling details are presented in Table 1. Two viruses, A/shoveler/Ubinskoe Lake/43/2017 (H12N5) (A/43) and A/teal/Chany/324/2017 (H12N5) (A/324), were isolated in the Novosibirsk region, Western Siberia, in September 2017; one virus (A/teal/Dagestan/1017/2018 (H12N5) (A/1017)) was isolated in the Dagestan Republic in January 2018; two viruses (A/shoveler/Novosibirsk region/999k/2018 (H12N5) (A/999) and A/mallard/Novosibirsk region/962k/2018 (H12N5) (A/962)) were isolated in the Novosibirsk region/Western Siberia in September 2018; and one virus (A/teal/Russia_Primorje/18-1377/2018 (H12N2) (A/1377-Amer)) was isolated in the Russian Far East in September 2018. The isolated strains are stored at the depository of CFTM. The 50% egg infectious dose (EID50) and the 50% tissue culture infectious dose (TCID50) for Madin–Darby Canine Kidney (MDCK) cells were determined for all these viruses, as previously described [12]. Virus titers were calculated by the Kerber method with Ashmarin–Vorobyov modification, as follows: log_10_TCID_50_/mL = lgDn − δ(ΣLi − 0.5).

### 2.2. Experimental Infection of Chickens and Mice

All animal experiments were conducted in biosafety level 3 facilities and were approved by the Ethics Committee of the Federal Research Center of Fundamental and Translational Medicine (No. 2019-3; data: 11 March 2019).

The intravenous pathogenicity index (IVPI) test for all isolated A/H12 viruses was performed and calculated according to the OIE standard protocol [13]. For this test, 10 six-week-old specific pathogen-free white Leghorn chickens were intravenously inoculated per virus, with 0.1 mL of 1:10 diluted infectious allantoic fluid (containing 10^6^EID_50_ of the virus). Clinical signs and mortality were monitored daily for 10 days. No unexpected deaths or clinical signs were observed. The pathogenicity index was calculated as the mean score per bird per observation. After 10 days, chickens were re-inoculated with 1 mL of the virus dilution containing 10^6^EID50 to obtain immune sera. On day 14, post-inoculation (p.i.) blood samples were collected from two chickens of each group and sera were harvested for antigenic analysis. 

To evaluate the pathogenicity of the viruses in mice, a group of 30 six-week-old BALB/c mice were lightly anesthetized and intranasally inoculated with 10^6^tissue culture infective doses (TCID_50_) of the virus in 50 µL of cell culture supernatant. A group of negative control mice was inoculated intranasally with 50 µL phosphate buffered saline (PBS). Animals were weighed and observed daily for 20 days post-infection (p.i.) for weight loss and clinical scores based on characteristics such as ruffled fur, hunched posture, and shivering. On day 21 p.i., all animals were euthanized, blood samples were collected, and serum was obtained. Serum samples were tested via a hemagglutination inhibition (HI) assay for the detection of antibodies against homologous viruses.

### 2.3. Antigenic Analysis

Antigenic analysis of A/H12 strains was performed by a hemagglutination inhibition (HI) test with chicken red blood cells using chicken polyclonal antisera obtained as described above. Before testing, all sera samples were heat-inactivated at 56 °C for 30 min. The HI test was performed according to the standard protocol (OIE). The highest dilution of the serum that completely inhibited hemagglutination was taken as HI [14]. Viruses were considered antigenically similar if their HI titer difference was no more than a two-fold dilution. 

### 2.4. Susceptibility to Neuraminidase Inhibitors

Neuraminidase activity and oseltamivir (F. Hoffmann–La Roche Ltd., Basel, Switzerland) susceptibility of the strains A/43, A/324, A/962, A/1377-Amer, A/1017, and A/999 were determined using a fluorescent neuraminidase inhibition (NAI) assay according to a previously described method. Briefly, viruses were standardized to an NA activity level 10-fold higher than that of the background, as measured by the production of a fluorescent product from 20-(4-methylumbelliferyl)-α-D-N-acetylneuraminic acid substrate (MUNANA; Sigma-Aldrich, Darmstadt, Germany). Drug susceptibility profiles were determined by the extent of NA inhibition after incubation with three-fold serial dilutions of NAIs. The 50% inhibitory concentrations (IC_50s_) were determined from the dose–response curve. The enzymatic reaction was read with a Varioskan Flash (Thermo Fisher Scientific, Waltham, MA, USA.) microplate reader with excitation and emission wavelengths of 360 and 460 nm, respectively. This work involved the use of equipment from the multiaccess center “Modern Optical Systems” of the Federal Research Center of Fundamental and Translational Medicine (Novosibirsk, Russia). All methods were performed in accordance with the relevant guidelines and regulations.

### 2.5. Sequencing 

RNA was isolated from cultured viral particles using a GeneJET viral DNA/RNA purification kit (Thermo Fisher Scientific, Waltham, MA, USA) and treated with TURBO DNase (Thermo Fisher Scientific, Waltham, MA, USA). Up to 200 ng of RNA was used for the DNA libraries, which were prepared using TruSeq RNA Sample Preparation Kit v2 (Illumina, San-Diege, CA, USA). Sequencing of the DNA libraries was conducted with a Reagent kit, Version 3 (600-cycle), on a MiSeq genome sequencer (Illumina) at the Siberian Branch of Russian Academy of Sciences, Genomics Core Facility (ICBFM SB RAS, Novosibirsk, Russia). Full-length genomes were assembled de novo with CLC Genomics Workbench version 9 (Qiagen, Hilden, Germany).

The genomes of two strains, A/shoveler/Novosibirsk region/999k/2018 and A/mallard/Novosibirsk region/962k/2018, were sequenced using next-generation sequencing, as previously reported by the National Institute of Animal Health, Tsukuba, Japan [15].

Nucleotide sequences of six H12 viruses have been deposited with the Global Initiative on Sharing All Influenza Data (GISAID) under the following numbers: EPI_ISL_331295 (A/teal/Chany/324/2017), EPI_ISL_331306 (A/shoveler/Ubinskoe_Lake/43/2017), EPI_ISL_331307 (A/teal/Dagestan/1017/2018), EPI_ISL_337402 (A/mallard/Novosibirsk region/962k/2018), EPI_ISL_337571 (A/shoveler/Novosibirsk region/999k/2018) and EPI_ISL_389024 (A/teal/Russia_Primorje/18-1377/2018).

### 2.6. Genetic Analysis

For phylogenetic analysis, we downloaded sequences of the H12 HA, N2, and N5 NA as well as internal genes from the GISAID database in May 2019. Sequences of the AIVs that the Federal Research Center of Fundamental and Translational Medicine possessed were aligned with the sequences downloaded from GISAID. After the alignment, sequences were used in the phylogenetic analysis performed using FastTree [16]. Sequences belonging to the clade involving Russian isolates in the present study were extracted from the first node, where the bootstrap value reached ≥95% from the periphery of the clade, and selected for further phylogeographic analyses.

Maximum likelihood phylogenetic trees were constructed with the abovementioned sequences, along with downloaded sequences after reducing the numbers by CD-HIT software [17] with a threshold of 98.5%. Tanglegrams were constructed from the pairs of trees obtained using Dendroscope 3 [18]. The taxa of six Russian isolates in adjacent trees were connected.

The location-annotated maximum clade credibility trees with the abovementioned sequences were constructed according to the Bayesian stochastic search variable selection by using Bayesian Evolutionary Analysis by Sampling Tree package version 1.8.4 [19]. Random walk models with a Bayesian statistical approach [20] and an uncorrelated relaxed clock model were applied for calculating Bayes factors in the present analysis. Then, the output tree was visualized by the spatial phylogenetic reconstruction of evolutionary dynamics using data-driven documents (SPreaD3) version 0.9.7 [21] with Bayes factors of 3.0 or more, which indicates stronger than moderate evidence [22] of viral dissemination.

Additional phylogenetic trees (for the visualization of phylogenetic relationships between Russian H12Nx and strains of different subtypes isolated worldwide) were built via MEGA 5 using the maximum likelihood method, utilizing the general time-reversible (GTR) nucleotide substitution model. Bootstrap support values were generated using 500 rapid bootstrap replicates. Comparative multiple amino acid sequence alignment and analysis was performed via BioLign 4.0.6. 

Detailed phylogenetic trees for each gene segment were generated using the Maximum likelihood estimation (ML)method and GTR + G substitution model. The robustness of each node was assessed by bootstrap method (500 replicates).

## 3. Results

### 3.1. Sampling and Virus Isolation 

Six influenza A/H12Nx viruses were isolated from wild migratory birds in autumn 2017 and 2018 in the context of annual influenza surveillance. All the viruses were isolated from the birds of the Anatidae family. Four of them (A/43, A/324, A/962, and A/999) were isolated in Western Siberia, one was isolated in the Caspian region (A/1017), and one was isolated in the Russian Far East (A/1377-Amer). Virus subtypes were determined based on the primary sequence of hemagglutinin and neuraminidase genes. Five viruses belonged to the H12N5 subtype, and one virus belonged to the H12N2 subtype. Sampling details are presented in Table 1.

### 3.2. Virological Characteristics

For all isolated A/H12 viruses, the 50% egg infectious dose (EID50) and the 50% tissue culture infectious dose (TCID50) were determined. All analyzed viruses efficiently replicated in embryonated chicken eggs (10^7.8^–10^8.3^EID50/mL) and MDCK cells (10^5.0^–10^5.8^50% TCID50/mL) in similar titers (Table 2). 

To determine the pathogenicity of A/H12 viruses for chickens, we intravenously inoculated six-week-old chickens with each virus, and IVPIs were calculated. All A/H12 viruses were low-pathogenic for chickens: all chickens survived and did not show any clinical signs of disease during the 10-day post-inoculation observation period (intravenous pathogenicity index = 0).

We also determined the pathogenicity of A/H12 viruses for mice. Mice did not show any clinical symptoms of the disease, such as body weight loss, ruffled fur, a hunched posture, or shivering. Mice post-infectious sera samples had no detectable levels of anti-HA antibodies on the 21st day p.i.

A fluorometric neuraminidase inhibition assay made it possible to examine the inhibition of neuraminidase activity by oseltamivir for the studied strains. The results show that all of the H12 strains isolated are sensitive to neuraminidase inhibitors (Table 2).

### 3.3. Antigenic Analysis

To determine the antigenic differences between A/H12 viruses, we performed antigenic analysis using the polyclonal chicken antisera raised against these viruses. All isolates demonstrated cross-reactivity with all chicken post-infectious antisera (Table 3). Only one isolate A/1377-Amer, which belonged to the American lineage, showed a four-fold reduction in HI titers for antisera raised against A/962 and A/999 as compared with homologous viruses, indicating the antigenic diversity between these viruses. 

### 3.4. Genetic Analysis

Six H12 Russian isolates were identified as five H12N5 AIVs (isolated in the Novosibirsk region, Ubinskoe Lake, Chany, and Dagestan) and one H12N2 AIV (isolated in the Primorsky region in the Russian Far East).

We analyzed each gene phylogenetically and examined proteins for the presence of specific amino acid substitutions. To visualize the general scheme of reassortment events, we constructed tanglegrams, and the taxa of six H12Nx isolates in adjacent trees were connected. We found that H12 HA genes of H12N5 AIVs were classified as the Eurasian lineage, and that of the H12N2 AIV was classified as the North American lineage (Figure 1). 

Five H12N5 AIVs isolated in Russia shared phylogenetically closely related H12 HA and N5 NA genes, and gene constellations of two H12N5 AIVs isolated in the Novosibirsk region (A/999 and A/962) were similar (Figure 1). The gene constellations of the other three H12N5 AIVs were different due to the reassortment (Figure 1). The H12 N2 AIV also possessed a distinct gene constellation. For detailed phylogenetic analysis, we constructed and considered each segment separately. 

#### 3.4.1. HA

HAs of strains from Western Siberia and Dagestan formed the common phylogenetic group of closely related sequences (Figure 2). This phylogenetic group clustered with HAs of strains mainly isolated in different parts of Eurasia: Western and Northern Europe, and Eastern Asia (Vietnam, Thailand, China, and Japan).

The HA of A/1377-Amer was phylogenetically different from the other Russian strains and related to the HA of the North American genetic lineage. The most similar HA sequences belonged to strains isolated both from the West Coast of America (California) as well as the East (Delaware Bay).

#### 3.4.2. NA

Sequences of the Russian H12N5 NA segments were similar to each other, belonging to the Eurasian genetic lineage (Figure 3), and were related to the NA of the strains isolated in Europe (the Netherlands, Poland, Sweden, and Croatia), Eastern Asia (Japan, China, and Korea), Southeastern Asia (Singapore), and Northeastern Asia (Kamchatka). Phylogenetic analysis, as well as the BLAST of the N2 segment, showed that the A/1377-Amer (H12N2) strain belongs to the Eurasian genetic lineage and is closely related to HxN2 strains isolated in the East Asian region, mainly in Japan, China, Korea and Vietnam (Appendix A).

#### 3.4.3. NP

The NP segment of the A/1377-Amer strain were found to belong to the North American genetic lineage and mainly related to the NP of AIVs isolated in California (Figure 4). The NPs of the other Russian H12Nx viruses were related to the AIV variants circulating in Eurasia (Eurasian genetic lineage). Strains of the Eurasian lineage were divided into two NP phylogenetic groups. The first included A/324 and A/1017 strains, related (according to phylogenetic tree and BLAST analysis) to strains primarily from Siberia (Chany Lake), the Black Sea region (Georgia), Europe (Kaliningrad), and Egypt. Only several strains with related NP were isolated in East Asia (Figure 4).

The second phylogenetic group, including the A/43, A/962, and A/999 strains, was mainly distributed in the Asian region (China, Japan, and Mongolia), although strains with similar NPs were also found in Europe (the Netherlands) and in the Arabian Sea region (Karachi, Pakistan) (Figure 4).

#### 3.4.4. NS

According to the phylogenetic analysis of the NS gene, all strains used to construct the phylogenetic dendrogram were divided into two main significantly distant groups of sequences (alleles A and B) [23]. North American and Eurasian genetic lineages were distinguished in each of these alleles. The NS genes of studied Russian H12N5 and H12N2 strains were genetically different: two strains (A/1017 and A/43) of six belonged to the allele B, and others (A/1377-Amer, A/324, A/962, and A/999) belonged to allele A (Figure 5). 

Allele B strains were similar in the NS sequences, belonging to the Eurasian genetic lineage, and were related to strains both from the Asian part of Eurasia (Bangladesh, Mongolia, and Japan) and from the European part (Czech Republic, Sweden, and the Netherlands).

NS allele A of Russian strains belonged to both genetic lineages. The NS gene of the A/1377-Amer strain belonged to the North American lineage, while the NS genes of A/324, A/962, and A/999 were similar to each other and belonged to the Eurasian lineage (related NSs were found in both European and Asian parts of Eurasia) (Figure 5).

#### 3.4.5. PA

The sequences of the Polymerase Acidic Protein (PA) segment of all studied Russian H12Nx strains belonged to the Eurasian genetic lineage (Figure 6). At the same time, they were different among themselves. In particular, A/43 was similar to the strains isolated in Siberia, Georgia, and Bangladesh. The PA segments of other strains belonged to another phylogenetic group and were similar to the PA segments of AIV variants, mainly those isolated in Japan. Moreover, strain A/1377-Amer contained PA segments that were more phylogenetically related to Japanese strains than strains from Siberia and Dagestan.

#### 3.4.6. PB1

Polymerase Basic Protein 1 (PB1) segments of all Russian H12Nx AIVs belonged to the Eurasian genetic lineage, and were subdivided into four phylogenetic clusters (Figure 7). 

PB1 segments of the A/962 and A/999 strains belonged to the subgroup (Hungary, Netherlands, Kamchatka, Mongolia, and China) maximally distant from other strains. The PB1 segment of A/1377-Amer belonged to the phylogenetic subgroup, which included strains from the Far East region of the Russian Federation, Japan, and China. The A/324 PB1 segment was also part of a separate phylogenetic subgroup and was similar to PB1 segments of the strains from the Asian part of Eurasia (Japan, Siberia, Mongolia, and Bangladesh). A/1017 and A/43 formed a phylogenetic group with strains from both the European part of Eurasia (Georgia) and the Asian part (Siberia, Mongolia, and Bangladesh) (Figure 7).

#### 3.4.7. PB2

According to the Polymerase Basic Protein 2 (PB2) segment phylogenetic analysis, Russian H12Nx strains were differentiated into three subgroups that belonged to the Eurasian genetic lineage (Figure 8). 

The Far Eastern strain, A/1377-Amer, was the most genetically distant from the other strains and was included in the phylogenetic subgroup with strains mainly from Japan. In addition, a PB2-related strain was isolated in 2014 in Alaska. The strains A/962 and A/999 were similar to each other and belonged to the phylogenetic subgroup formed by strains that were isolated in different regions of Eurasia: Europe (the Netherlands), North Asia (Siberia), East Asia (Japan, China, and Mongolia), and South Asia (Bangladesh). A/1017, A/324, and A/43 were similar to each other (first two were more closely related). These strains formed the common phylogenetic subgroup with strains from Siberia (northern Asia), the Netherlands (Western Europe), Croatia (central Europe), Kamchatka (northeast Eurasia), and Korea (eastern Asia).

#### 3.4.8. MP

Matrix Protein (MP) segments of all Russian H12Nx AIVs belong to Eurasian genetic lineage, and subdivided into several phylogenetic subgroups (Figure 9). 

The MP of strain from the Russian Far East (A/1377-Amer) belonged to the phylogenetic subgroup formed by strains that were isolated Japan and China. MP of A/324 belongs to the separate phylogenetic subgroup of MP from the Asian part of Eurasia (Japan, Mongolia, Korea and China). The MP of other Russian H12N5 strains forms a phylogenetic subgroup of viruses mainly isolated in the Asian part of Eurasia (European part of Eurasia represented only by MP from Georgia). Thus, influenza viruses with an MP segment similar to the MP of Russian H12Nx strains predominantly circulate in Asia, but their penetration into Europe is possible (for example, through Dagestan and Georgia).

Additionally, we constructed five eight-segment-merged maps based on phylogeographic analyses for each segment (Appendix A). The results also showed that A/1377-Amer takes its HA, NP and NS segments from the American lineage (closest relatives isolated in 2016–2017 in California, Delaware, Alberta), while the five other segments likely originate from viruses in the Far-East Region (closest relatives isolated in Japan, 2016 to 2017). The geographical distribution of the NS segments (A or B allele) suggests that the closest viruses of the B allele were isolated since 2010 across Eurasia (Sweden to Bangladesh), while the closest viruses to the A allele correspond to recent HxNy isolates in neighboring areas (Chany, Hungary, Georgia, Mongolia since mainly 2015). The genetic connection of the AIVs isolated from the Novosibirsk region to the AIVs across the Eurasian continent could be explained through the overlap in multiple migration flyways, including the Black Sea–Mediterranean Flyway, East Africa–Western Asia Flyway, Central Asia Flyway, and East Asia–Australia Flyway in the Novosibirsk region (Appendix A).

Genetic analysis of the HA amino acid sequences of the investigated H12Nx strains showed the presence of monobasic cleavage sites: VPQVQDR*GL for A/324, A/43 and A/1017 strains; VPQVQNR*GL for A/962 and A/999; VPQVQSR*GL for A/1377-Amer. Monobasic cleavage sites are characteristic of LPAI (low-pathogenic avian influenza) viruses [24].

Around the receptor-binding site (RBS), HAs of investigated strains contain amino acid residues characteristic of the H12 subtypes [25] Y98, W153, T155, H183, 190E, 194L, and 195Y in the RBS; R224, G225, Q226, Q227, G228, and R229 in the left edge of the RBS; and G134, T135, S136, K137, and A138 in the right edge of the RBS (the H3 amino acid numbering used).

The viruses are characterized by their avian receptor-binding specificity (via the presence of Q and G at positions 226 and 228 of HA, H3 numbering) [26] according to Bui and co-authors [27].

Amino acid substitutions that were previously described in the literature as significant were found in internal genes (PA, PB1, and PB2). PA gene segments of all strains in the study had S149 and A515 amino acids; PB1 had L598; and PB2 had V553, E391, and E627, suggesting the difference of replication activity of the virus in different models. In total, according to the amino acid analysis of all Russian H12Nx strains, they contain mutations that, according to published data, can affect the biological properties of viruses. Part of the amino acid substitutions detected in Russian strains can increase the virulence of the virus, while others can reduce virulence. Taken together, and in combination with the main marker, the monobasic HA cleavage site, this results in the low-pathogenicity of the investigated strains (LPAI) that was confirmed in experiments on model animals (Table 4). Additionally we showed amino acid differences between investigated strains (Appendix A).

## 4. Discussion

Wild birds are known to be the main reservoir for a wide variety of virus lineages and subtypes. Information about the H12 subtype virus is limited due to its rare isolation in nature [6]. In the 1990s, there was a hypothesis that ducks very weakly support the H12 subtype [7], and even that this subtype had disappeared and that this “old subtype” may be endangered by influenza reservoirs in nature [36].

A comprehensive study was conducted by Wille et al. (2018) to summarize data about the evolution and ecology of H12 AIVs using long-term surveillance from 2002 to 2009 [6]. Considering our results, we agree with the statement about “the H12 enigma” [6] because we have not found the frequent isolation of H12 over 10 years of active monitoring of waterfowl in Russia. In our study, during surveillance, we identified only six viruses in the Northern part of Eurasia, in Russia. A subtype combination of five strains was H12N5, which was the most common for H12 viruses, as was shown in previous studies. One virus was H12N2, and there are only three sequences available for this combination (GISAID). The isolation rate of the H12 subtype was 0.36% in 1652 samples collected in 2017–2018, while the total AIV isolation rate was 4.18%. We found that one studied H12 virus is an inter-continental reassortant: three of the genome segments belong to North American viruses and the other five belong to Eurasian viruses, confirming the hypothesis that this rare subtype circulates in the general genetic pool of viruses and is involved in the process of reassortment. In total, the sequences of only 349 avian H12Nx strains isolated in three continental regions are represented in GISAID: a total of 274 in North America, 33 in Asia, and 31 in Europe. Additionally, six H12 strains are known to have been isolated from the environment. The majority of characterized H12 viruses were isolated in North America and contained segments of American lineages, while Eurasian H12 viruses usually contain Eurasian avian-like segments. Some viruses have been found in South America, but the complete genomes are mostly unknown.

In spite of the distant sites at which our five H12N5 Eurasian viruses were isolated, their HAs form a tight cluster of highly homologous sequences. The most closely related sequences corresponded to H12Nx viruses that were isolated across Eurasia, from Netherlands to Vietnam and Japan since 2010. This might suggest that the Eurasian H12 segment is relatively homogeneous in sequence (with slow or neutral evolution) and that its circulation area spreads across the whole Eurasian region (despite the relatively low isolation rate from ducks). A similar conclusion could be applied to the American H12 segment, since the closest sequences of the H12 segment of A/1377 are found in viruses isolated in California and Delaware in 2015–2016. 

Furthermore, the NAs of the five H12N5 viruses isolated in west Siberia and the Caspian form a very tight cluster. The closest relative sequences belong to HxN5 viruses that were isolated across Eurasia (from Kamchatka to Germany and Italy) since 2013. The tight clustering of both the H12 sequences and the N5 sequences may perhaps suggest that the fitness of the virus relies on some preferred association/combination of the two viral glycoproteins. As for the other six viral genomic segments, it should be emphasized that the two isolates A/962 and A/999 (both isolated in the west Siberia region) share the same genomic constellation and could be the same virus pool in sample site.

The central question raised by Wille et al. (2018) [6] was as follows: are ducks the major reservoir of the H12 subtype, or do we need to discover other important host species? 

Currently, the database of 349 avian H12 strains includes 200 isolated from ducks and 115 isolated from shorebirds, mostly from sandpipers. In our study, we mainly focused on duck reservoir surveillance and confirmed the statement about the relatively low rate of H12 isolation from ducks. Our previous studies did not report the isolation of H12 viruses from more than 25000 wild birds during a 10-year surveillance program [8,37] in a very wide territory from the Caspian Sea to Pacific Ocean. Our present data are consistent with the data provided by Muzyka et al. who collected 6281 samples from wild birds representing 27 families in the Azov, Black Sea region and also isolated only one H12N8 virus (in a period from 2001 to 2012) [38].

All our viruses were isolated from dabbling ducks of three species (*Anas crecca, A. clypeata, A. platyrhynchos*) which are the most numerous species of dabbling ducks and Anseriformes in Northern Eurasia, covering the region from Europe to Kamchatka and Japan. Combined with the published data, this suggests that diving ducks do not seem to be the primary reservoir of the H12 subtype in Northern Eurasia. There seems to be no strong host-correlation with the H12Nx subtype, as shown for H13 and H16 gull-like viruses [39,40] or H9N2 viruses associated with Galliformes, based mainly on studies with chickens and turkeys [41]. No strong species effect was shown to be associated with virus diversity, similar to the results described previously for North American AIV [42].

All birds from which viruses are isolated are long-distance migratory birds. All of the H12 isolation sites in this study can combine significant populations of waterfowl from different places within the migration routes of their territories ([43], Appendix A). The reassortant A/1377-Amer isolation site is located in the Far East, Primorye, which is the part of the Far Eastern and East Siberian territorial groupings of birds connected with the East Asian–Australasian Flyway [44], while the isolation points of A/324, A/962, A/999, and A/43 viruses are located in the south of Western Siberia at the intersection of three flyways: Central Asian Flyway, Asian–East African Flyway, and Black Sea–Mediterranean Flyway [43,45]. 

The western part of the Caspian Sea where the A/1017 virus was isolated is located in the area of the intersection of two flyways: the Asian–East African Flyway and Black Sea–Mediterranean Flyway (Appendix A, [45]). Therefore, taking into account the location of the H12 virus isolation points in Siberia and the Caspian Sea, we can use our maps and phylogenetic tree analysis to suggest associations between the studied segments and other phylogenetically similar gene pools of viruses which were isolated in Egypt, the Caucasus (Georgia), Southern and Central Europe, the Balkans, Mongolia, and China. Our findings indicate a mixing in the Siberia region of various genetic variants of the AIV circulating in Eurasia. One possible explanation for this is the spreading of different LPAI viruses to Siberia acting as an “LPAI virus hub” through different routes, followed by mixing with possible reassortment events [8].

Fortunately, in the studied strains, we did not observe segments strongly associated with highly pathogenic avian influenza viruses nor human or mammal viruses that could lead to a high risk of active transmission (such as H5Nx and H7Nx). However, according to the phylogenetic analysis, we identified genomic segments of Russian H12Nx related to the segments that belonged to some HPAI strains. For example, in the PB2 segment, the A/962 and A/999 strains are similar to the H5N8 A/great crested grebe/Uvs Nuur Lake/341/2016_A/H5N8 strain, as well as to the H7N1 A/duck/Bangladesh/24705/2015 subtype strain. Moreover, the PB2 and PB1 of strains A/962 and A/999 are phylogenetically related to PB2 and PB1 of H5N8 and H5N5 HPAI (highly pathogenic avian influenza) strains. The sequences of the PB2 segment of the A/324, A/1017, and A/43 strains belong to the same phylogenetic group as the PB2 of the HPAI H5N5. According to the phylogenetic analysis, the NP genes of A/324 and A/1017 strains are closely related to HPAI H5N8. Therefore, the example of Russian H12Nx strains shows that a possible exchange of genome segments occurred between low-pathogenic and highly pathogenic variants of the influenza virus. In addition, HPAI and LPAI circulate jointly or intersect in places where birds congregate, which makes it possible for HPAI to spread by the same birds and routes (migratory pathways) through which LPAI spreads. It has been shown that H12Nx viruses have a specific gene pool with frequent reassortment events. We confirmed this fact in our research.

This information is important for the annual screening of the characteristics of current viruses, especially those relatively rare subtypes that probably have a minor reservoir and still need to be studied in the future.

## 5. Conclusions

During surveillance, we isolated AIV subtypes with low prevalence (five H12N5 and one H12N2 viruses) in three different distinct geographic regions of Northern Eurasia (Russia). H12N2 from the Far East region was a double reassortant containing HA, NS, and NP segments of the American lineage and others from the classical Eurasian avian-like lineage. H12N5 viruses contained all Eurasian lineage segments.

We have suggested a phylogeographical scheme of reassortment events associated with geographical groups of aquatic birds and their migration flyways. The H12N5 strain is of particular interest, as this virus has been found in common teal in the Russian Far East region, and its three segments are strongly related to the North American AIV lineages and clearly demonstrates that viral exchange between the two continents does occur. 

## Figures and Tables

**Figure 1 microorganisms-07-00643-f001:**
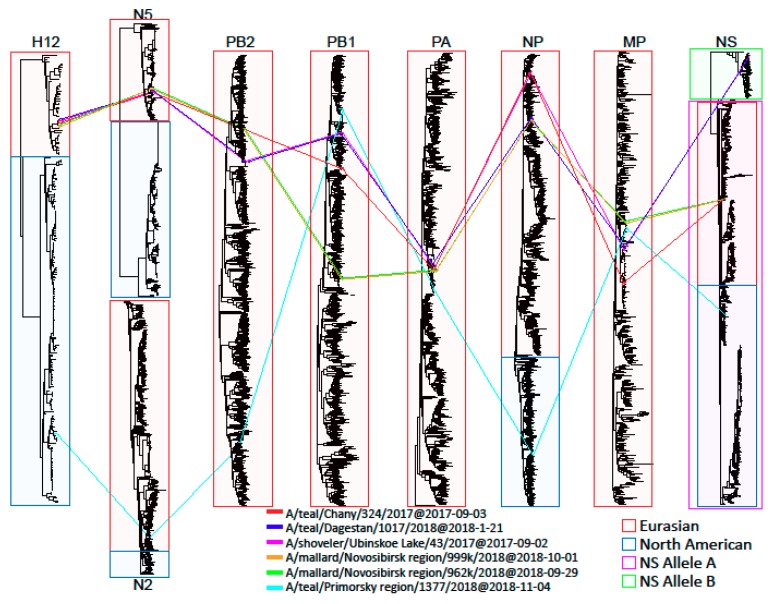
Tanglegrams showing relationships between H12Nx viruses.

**Figure 2 microorganisms-07-00643-f002:**
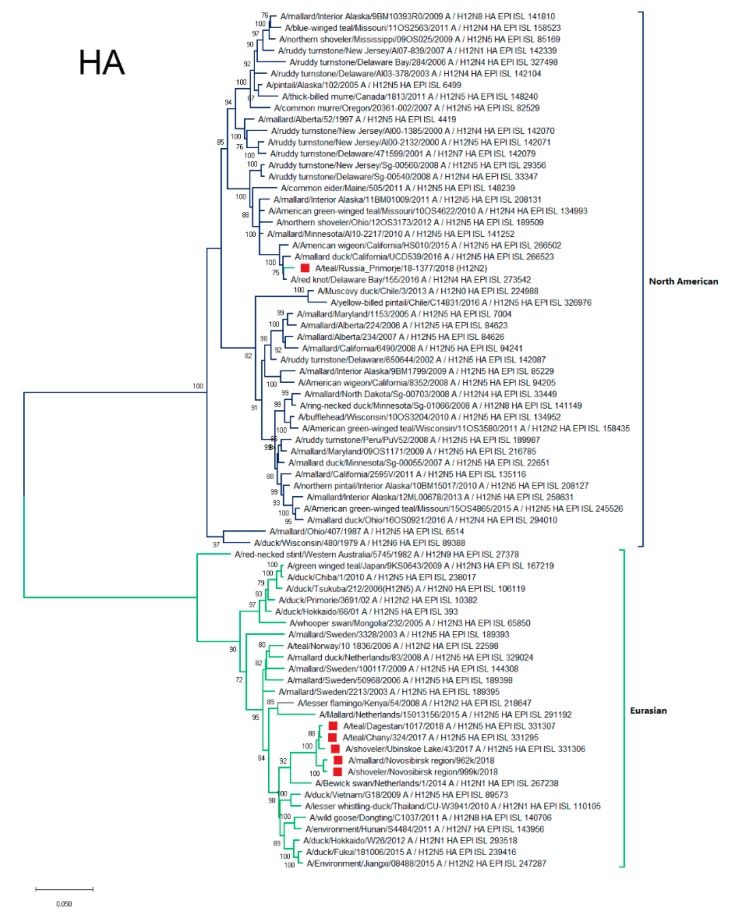
Phylogenetic tree of the hemagglutinin (HA) gene of Avian Influenza Viruses. The tree was constructed using MEGA X software with the Maximum likelihood estimation algorithm (general time reversible (GTR)+G model) and bootstrap analysis with 500 iterations. Russian H12N2 and H12N5 viruses are indicated using red squares.

**Figure 3 microorganisms-07-00643-f003:**
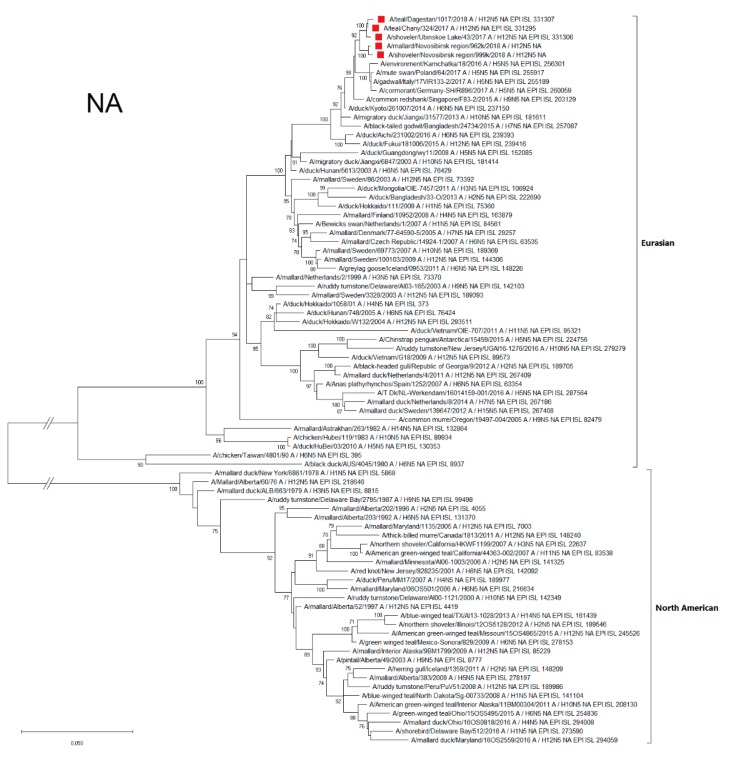
Phylogenetic tree of the neuraminidase (NA) gene of AIVs. The tree was constructed using MEGA X software with the ML algorithm (GTR+G model) and bootstrap analysis with 500 iterations. Russian H12N5 viruses are indicated using red squares.

**Figure 4 microorganisms-07-00643-f004:**
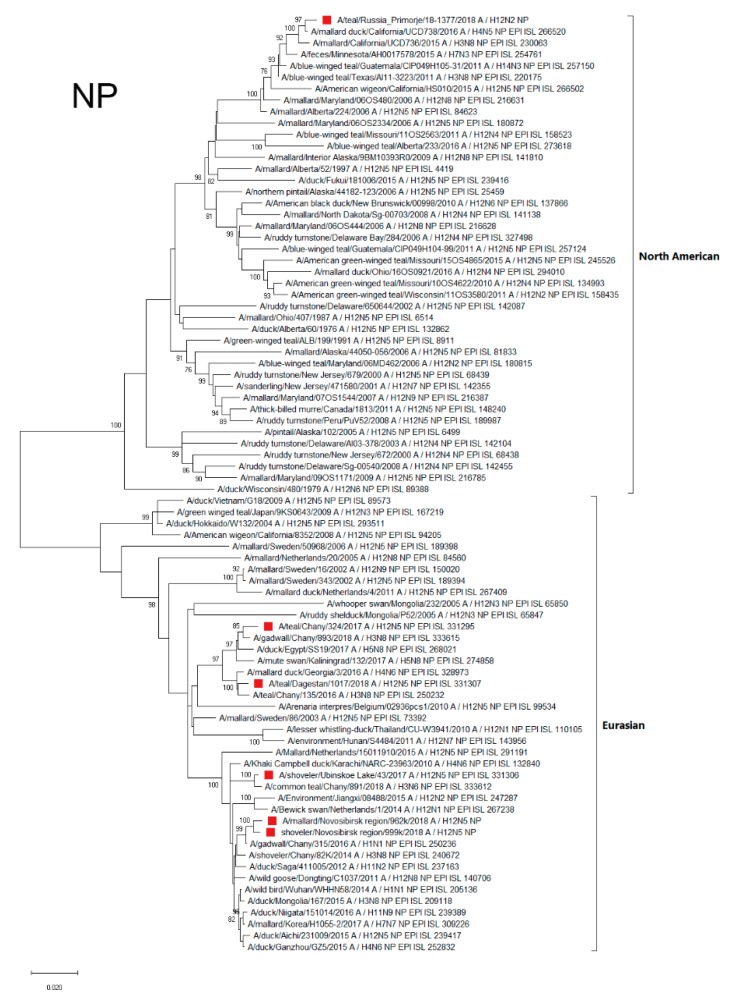
Phylogenetic tree of the nucleoprotein (NP) gene of AIVs. The tree was constructed using MEGA X software with the ML algorithm (GTR+G model) and bootstrap analysis with 500 iterations. Russian H12N2 and H12N5 viruses are indicated using red squares.

**Figure 5 microorganisms-07-00643-f005:**
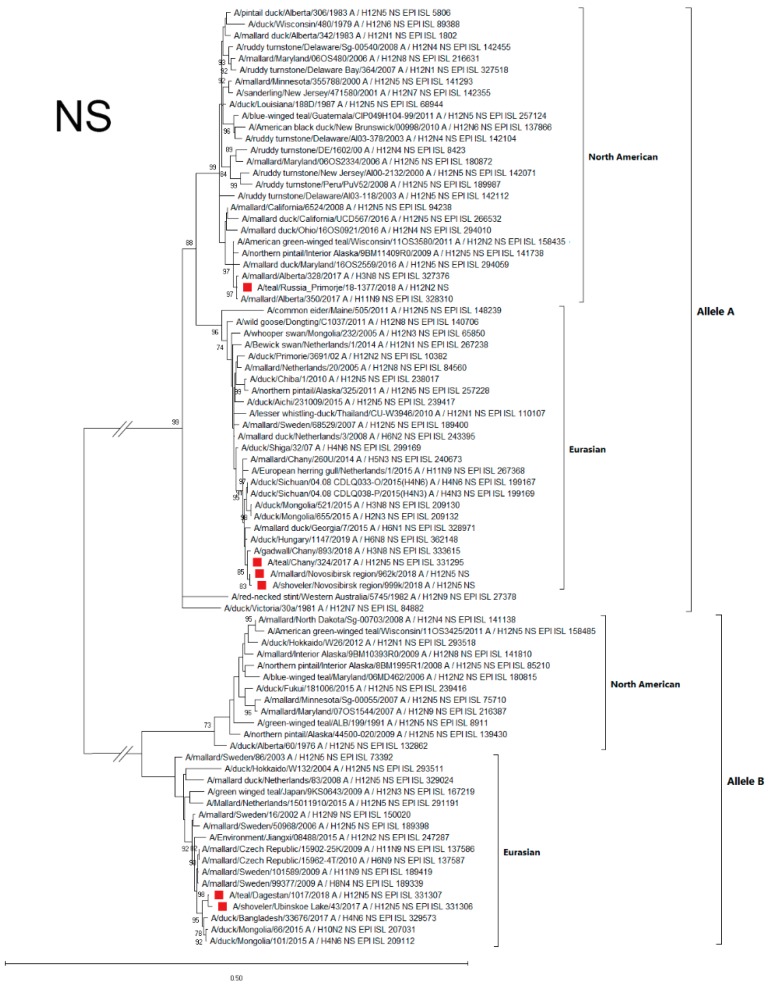
Phylogenetic tree of the non-structural (NS) gene of AIVs. The tree was constructed using MEGA X software with the ML algorithm (GTR+G model) and bootstrap analysis with 500 iterations. Russian H12N2 and H12N5 viruses are indicated using red squares.

**Figure 6 microorganisms-07-00643-f006:**
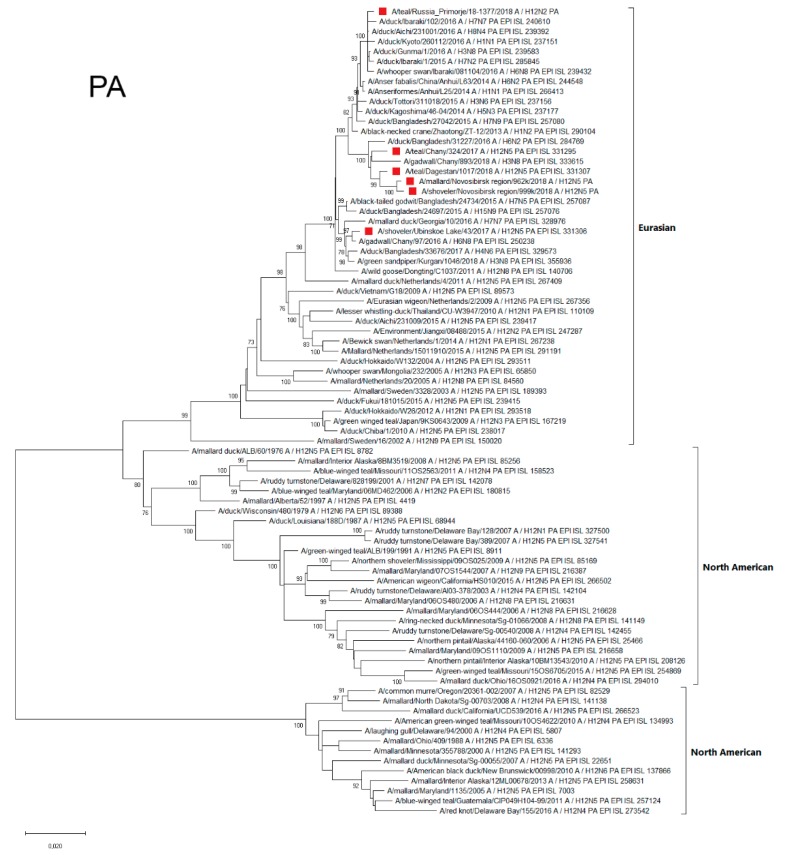
Phylogenetic tree of the PA gene of AIVs. The tree was constructed using MEGA X software with the ML algorithm (GTR+G model) and bootstrap analysis with 500 iterations. Russian H12N2 and H12N5 viruses are indicated using red squares.

**Figure 7 microorganisms-07-00643-f007:**
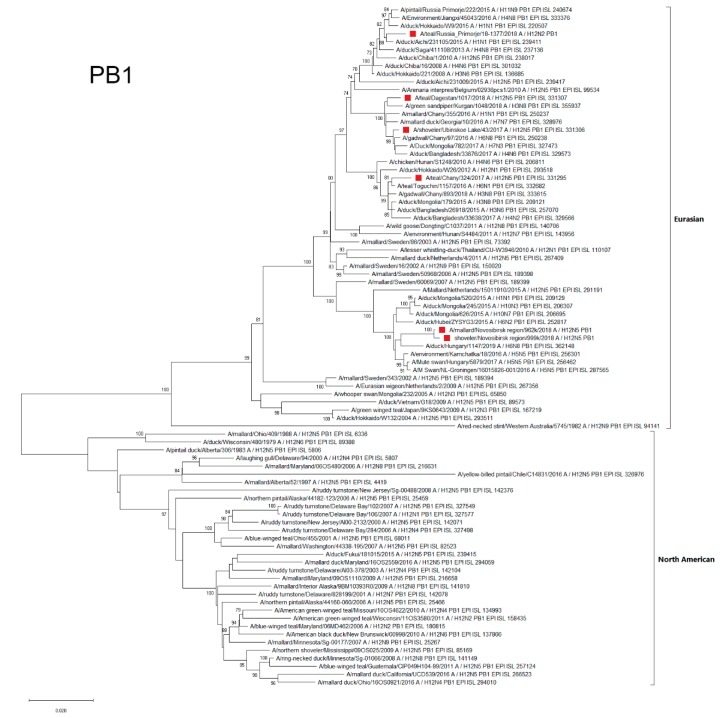
Phylogenetic tree of the PB1 gene of AIVs. The tree was constructed using MEGA X software with the ML algorithm (GTR+G model) and bootstrap analysis with 500 iterations. Russian H12N2 and H12N5 viruses are indicated using red squares.

**Figure 8 microorganisms-07-00643-f008:**
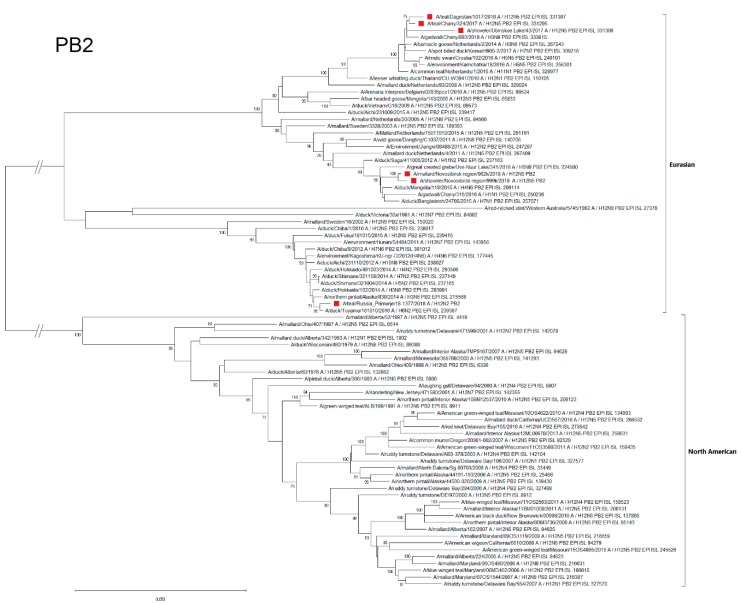
Phylogenetic tree of the PB2 gene of AIVs. The tree was constructed using MEGA X software with the ML algorithm (GTR+G model) and bootstrap analysis with 500 iterations. Russian H12N2 and H12N5 viruses are indicated using red squares.

**Figure 9 microorganisms-07-00643-f009:**
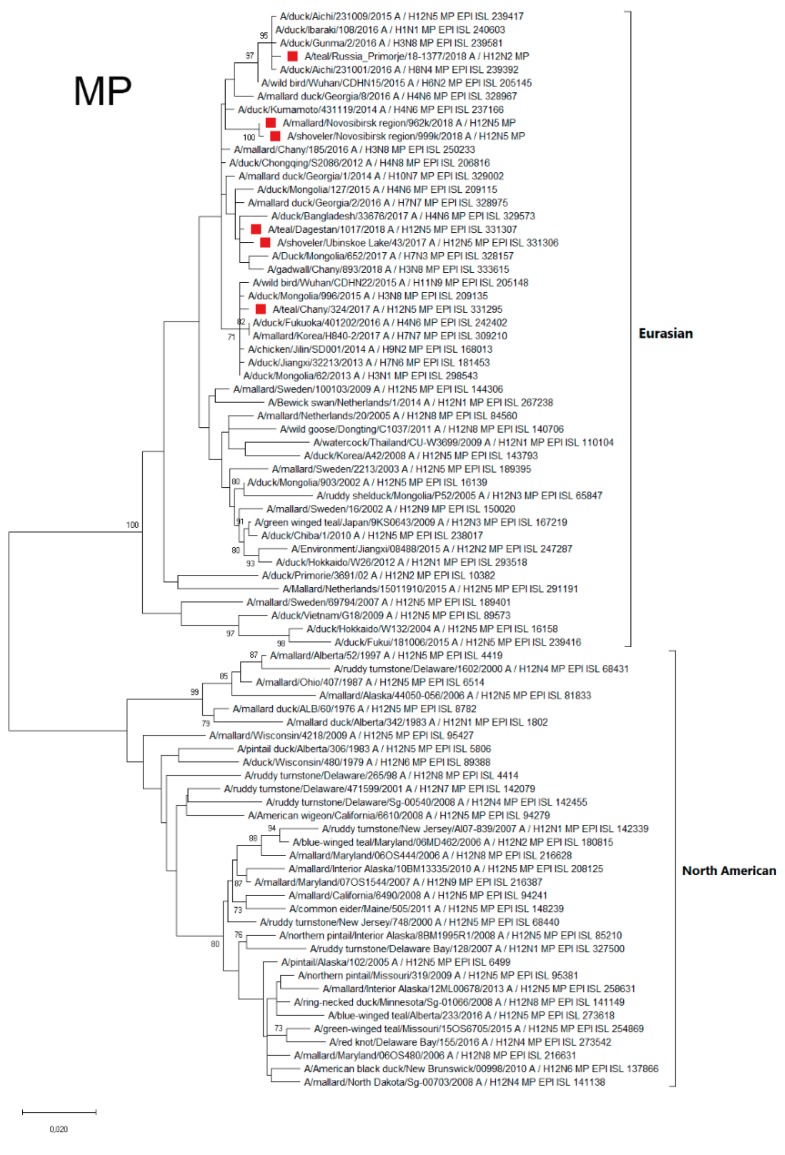
Phylogenetic tree of the MP gene of AIVs. The tree was constructed using MEGA X software with the ML algorithm (GTR+G model) and bootstrap analysis with 500 iterations. Russian H12N2 and H12N5 viruses are indicated using red squares.

**Table 1 microorganisms-07-00643-t001:** Sampling details and viruses.

Viruses	Sampling Date	Sampling Region/ Sample Size at Site	Host Species (Latin Name)	Host Species (English Name)	Virus Subtype	Strain Name
A/43	September 17	Western Siberia/590	*Anas clypeata*	Shoveler	H12N5	A/shoveler/Ubinskoe Lake/43/2017
A/324	September 17		*Anas crecca*	Common teal	H12N5	A/teal/Chany/324/2017
A/1017	January18	Caspian region/304	*Anas crecca*	Common teal	H12N5	A/teal/Dagestan/1017/2018
A/962	September 18	Western Siberia/478	*Anas platyrhynchos*	Mallard	H12N5	A/mallard/Novosibirsk region/962k/2018
A/999	September 18		*Anas clypeata*	Shoveler	H12N5	A/shoveler/Novosibirsk region/999k/2018
A/1377-Amer	November 18	Far East/280	*Anas crecca*	Common teal	H12N2	A/teal/Russia_Primorje/18-1377/2018

**Table 2 microorganisms-07-00643-t002:** H12Nx virus characteristics.

Viruses	log_10_TCID_50_/mL	log_10_EID_50_/mL	IVPI	Pathogenicity for Mice	Oseltamivir Carboxylate IC_50_ (nM)	Phenotype ^b^
A/43	5.4 ± 0.3	8.3 ± 0.3	0	np ^a^	12.47	RI
A/324	5.6 ± 0.3	8.3 ± 0.2	0	np	7.5	S
A/1017	5.0 ± 0.2	7.9 ± 0.4	0	np	9.2	S
A/962	5.4 ± 0.2	8.0 ± 0.2	0	np	0.4	S
A/999	5.8 ± 0.3	7.8 ± 0.4	0	np	0.4	S
A/1377	5.3 ± 0.2	8.0 ± 0.3	0	np	4.4	S

Note: ^a^ np = non-pathogenic; ^b^ the phenotype of susceptibility to neuraminidase inhibitions (NAIs) according to WHO guidelines: S, susceptibility or normal inhibition (<10-fold increase in IC50 over Ca/09); RI, reduced inhibition (10- to 100-fold increase in IC50 over Ca/09); Ca/09, vaccine strain A/California/07/2009(H1N1) pdm09 that was isolated in the pandemic period and demonstrated normal inhibition by oseltamivir. TCID: 50% tissue culture infectious dose; IVPI: intravenous pathogenicity index.

**Table 3 microorganisms-07-00643-t003:** Hemagglutination inhibition (HI) antigenic analysis of influenza A/H12Nx viruses.

Chicken Post Infectious Sera	Antigens					
	A/43	A/324	A/1017	A/962	A/999	A/1377
A/43	160	160	80	160	320	80
A/324	160	160	80	160	640	160
A/1017	160	160	160	320	640	160
A/962	80	80	80	160	160	40
A/999	80	80	80	80	160	40
A/1377-Amer	80	80	80	80	160	160

Note: titers of homologous serum and antigen are marked grey.

**Table 4 microorganisms-07-00643-t004:** Amino acid substitutions of Russian H12N2 and H12N5 strains associated with replication, pathogenicity, and transmission.

Gene	Amino Acid Site	Strain	Effect	Subtype Showed to Be Affected	Reference
PA	149S	A/1377, A/324, A/962, A/999, A/1017, A/43	P149S—limited lethality in mice	H5N1	[28]
PA	515A	A/1377, A/324, A/962, A/999, A/1017, A/43	T515A—polymerase activity decreasing	H5N1	[29]
PB1	598L	A/1377, A/324, A/962, A/999, A/1017, A/43	P598L—replication decreasing in MDCK	H1N1, H5N1	[30]
PB1-F2	66S	A/962, A/999	N66S—replication increasing	H5N1	[31]
PB2	553V	A/324	I553V—polymerase activity decreasing	H5N1	[28]
PB2	391E, 627E	A/1377, A/324, A/962, A/999, A/1017, A/43	Q391E—virulence decreasing in ferrets; K627E—replication decreasing in mammalian cells	H5N1	[32,33]
PB2	701D	A/1377, A/324, A/962, A/999, A/1017	N701D—lethality increasing in mice	H5N1	[34]
PB2	89V, 309D, 339K, 477G, 495V, 627E, 676T ^1^	A/1377, A/324, A/962, A/999, A/1017, A/43	L89V, G309D, T339K, R477G, I495V, K627E, A676T—polymerase activity increasing in mouse cells	H5N1	[35]

^1^ Except for the A/43 strain.

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
