# Peer review of "Characterization and Phylodynamics of Reassortant H12Nx Viruses in Northern Eurasia"

_microorganisms, 2019, doi:10.3390/microorganisms7120643_

Round 1

Reviewer 1 Report

In this manuscript by Kirill Sharshov et al., the authors genetically and phenotypically characterized six H12Nx avian influenza viruses that were isolated over a ten year period in northern Eurasia. This led the authors to propose a phylogeographical scheme for reassortment events, taking account of the migration flyways. (I reviewed the previous version, manuscript 603123)

The main findings are:

Five H12N5 and one H12N2 avian influenza viruses were isolated from wild birds of the Anatidae family in 2017 and 2018 in Northern Eurasia. Their full genomes were sequenced. They all showed a Low Pathogenicity phenotytpe, as shown by the Intra Venous Pathogenicity Index (IVPI=0). Consistent with that, the HA sequences have a monobasic cleavage site, as expected, and they show a specificity towards avian receptor. They also showed no pathogenicity for mice (after intra-nasal inoculation with 10^6 TCID50). They were all sensitive to the neuraminidase inhibitor oseltamivir.-          The H12 hemaglutinin of five of the six viruses were antigenically similar, as indicated by the cross-reactions in Hemaglutination Inhibition tests. However, the HA of A/1377 (i.e. the H12N2 virus from Russia far east) was antigenically distant, as revealed by a fourfold reduction in HI titer. The genomic constellations of the five H12N5 viruses were very similar, with their eight segments belonging to the Eurasian lineage. However, two among these five H12N5 (A/1017 and A/43) distinctively harboured a NS segment of allele B (Figure 1). The genomic constellation of A/1377 (the H12N2 virus) was clearly distinct, with three segments (H12, NP and NS) belonging to the North American lineage and the five remaining segments closely related to segments from Japanese viruses.

The manuscript contains interesting information. However, the informations and their interpretations should be more synthetic. The discussion if much too long. Below are a number of points that need to be addressed by the authors.

Major remarks

Lines 54-55. There is no High Pathogenicity Avian Influenza virus in the subtypes H2, H4, H8 and H14 (and refs [3-5] are relative to only H5 and H7 viruses). 

There is no Figure 1 in the manuscript (only its title). 

Lines 245-48. In spite of the distant sites at which the five viruses were isolated, their HAs form a tight cluster of highly homologous sequences. The most closely related sequences corresponded to H12Nx viruses that were isolated across Eurasia, from Netherlands to Vietnam and Japan since ~2010. This might suggest that the Eurasian H12 segment is relatively homogeneous in sequence (with slow or neutral evolution) and that its circulation area spreads across the whole eurasian region (even if it is rarely isolated).

A similar interpretation may apply to the American H12 segment, since the closest relatives of the H12 segment of A/1377 are found in viruses isolated in California and Delaware in 2015-2016. 

Lines 258-264. Again, the NAs of the five viruses isolated in west-Siberia and Caspian form a very tight cluster. The closest relative sequences belong to HxN5 viruses that were isolated across Eurasia (from Kamtchatka to Germany and Italy) since 2013. Further, the tight clustering of both the HA12 sequences and the N5 sequences may perhaps suggest that the fitness of the virus relies on some preferred association of the two viral glycoproteins. 

As for the other six viral genomic segments, the authors should emphasize the fact that the isolates 962 and 999 (both isolated in the west-Siberia region) share the same genomic constellation. 

Fig 10a is unnecessary and confusing. It is sufficient, and more informative, to say that 1377 has its HA, NP and NS segments from the American lineage (closest relatives isolated in 2016-2017 in California, Delaware, Alberta), while the five other segments likely originate from viruses in the far-east region (closest relatives isolated in Japan, 2016 to 2017). 

Likewise, Figs 10b to 10e are confusing and poorly informative. Firstly, the geographical distribution of the HA and NA segments was already described (see above my comments to lines 245-264). The geographical distribution of the NS segments (A or B allele) is also easily described: closest relatives of the B allele were isolated since ~2010 across Eurasia (Sweden to Bangladesh), while closest relatives to the A allele correspond to recent HxNy isolates in neighboring areas (Chany, Hungary, Georgia, Mongolia since ~2015). As regards the five remaining segments of these “four” viruses (962 and 999 sharing the same constellation), similar interpretations could be made with no need of these confusing maps. 

Lines 406-408. Why do the authors state that the given amino acids in the polymerase segments suggest the low replication of the virus? Further, the following lines (408-414) do not provide a satisfactory explanation. And it is obvious that these H12 viruses are Low Pathogenicity Avian Influenza viruses, not only from the experimental infection of chickens, but also from the fact that a multibasic HA cleavage site has never been observed in viruses other than H5 or H7. 

The discussion section is much too long, unnecessarily complicated, and contains several redundant paragraphs (for exemple lines 439-40 are repeated in lines 514-19). It would be sufficient, and more informative, to discuss the main points (above in this review) in a synthetic manner. 

Lines 439-442.   “… is an inter-continental reassortant..”. Further, “active” and “actively” are excessive in the statement  “this rare subtype actively circulates...and is actively involved…”. Further again, it is not that particular subtype (i.e. the H12N2 subtype), but rather the H12 genomic segment.

Lines 453-467.  Fig S1. The alignment is somewhat confusing. Firstly, the colors here are useless. Secondly, the sequences should be grouped by similarity (i.e A/43 and A/1017 grouped; 1377 and 324 grouped). Thirdly, it would be more readable with one consensus sequence, and the six sequences aligned to the consensus (rather than choosing A/1377 as the reference sequence).  

Minor remarks

Column 3 in table 2 is unnecessary (the values correspond to [column4 – column2]).

Line 468. …long-distance migratory birds..(instead of distant migrants)

Author Response

Dear Reviewer,

Thank you for your useful comments and questions which helped to improve the manuscript.

Please see below each point for a detailed response on how we incorporated your feedback into the manuscript.

Major remarks

Lines 54-55. There is no High Pathogenicity Avian Influenza virus in the subtypes H2, H4, H8 and H14 (and refs [3-5] are relative to only H5 and H7 viruses).

Response

We missed the reference of PNAS. We agree there is no HPAI exceptH5/H7  However, the avian influenza virus subtypes H2, H4, H8, and H14 could support a highly pathogenic phenotype as have been shown experimentally after introduction of the polybasic cleavage site into the HA. We corrected and added reference [5 -Veits J, Weber S, Stech O, Breithaupt A, Gräber M, Gohrbandt S, Bogs J, Hundt J, Teifke JP, Mettenleiter TC, Stech J.Avian influenza virus hemagglutinins H2, H4, H8, and H14 support a highly pathogenic phenotype. Proceedings of the National Academy of Sciences of the United States of America 2012, 109, 2579-2584.]. Therefore, monitoring and investigation of the pathogenic potential of new and rare subtypes is important for the seeking and evaluating the pathogenic potential of new virus variants.

 There is no Figure 1 in the manuscript (only its title).

Response

We insert the Figure 1

 Lines 245-48. In spite of the distant sites at which the five viruses were isolated, their HAs form a tight cluster of highly homologous sequences. The most closely related sequences corresponded to H12Nx viruses that were isolated across Eurasia, from Netherlands to Vietnam and Japan since ~2010. This might suggest that the Eurasian H12 segment is relatively homogeneous in sequence (with slow or neutral evolution) and that its circulation area spreads across the whole Eurasian region (even if it is rarely isolated).

 Response

We completely agree with Reviewer concerning this important statement. We included this in Discussion section. We are very grateful for this addition.

A similar interpretation may apply to the American H12 segment, since the closest relatives of the H12 segment of A/1377 are found in viruses isolated in California and Delaware in 2015-2016.

Response

Again we completely agree with Reviewer. We included this in Discussion section. Thanks a lot for useful comments. Please see the marked parts.

 Lines 258-264. Again, the NAs of the five viruses isolated in west-Siberia and Caspian form a very tight cluster. The closest relative sequences belong to HxN5 viruses that were isolated across Eurasia (from Kamtchatka to Germany and Italy) since 2013. Further, the tight clustering of both the HA12 sequences and the N5 sequences may perhaps suggest that the fitness of the virus relies on some preferred association of the two viral glycoproteins.

As for the other six viral genomic segments, the authors should emphasize the fact that the isolates 962 and 999 (both isolated in the west-Siberia region) share the same genomic constellation.

Response

We included the highlighting of the similar genomic constellation. Thank you very much! We used this comment to correct and improve the Discussion section

 Fig 10a is unnecessary and confusing. It is sufficient, and more informative, to say that 1377 has its HA, NP and NS segments from the American lineage (closest relatives isolated in 2016-2017 in California, Delaware, Alberta), while the five other segments likely originate from viruses in the far-east region (closest relatives isolated in Japan, 2016 to 2017).

Likewise, Figs 10b to 10e are confusing and poorly informative. Firstly, the geographical distribution of the HA and NA segments was already described (see above my comments to lines 245-264). The geographical distribution of the NS segments (A or B allele) is also easily described: closest relatives of the B allele were isolated since ~2010 across Eurasia (Sweden to Bangladesh), while closest relatives to the A allele correspond to recent HxNy isolates in neighboring areas (Chany, Hungary, Georgia, Mongolia since ~2015). As regards the five remaining segments of these “four” viruses (962 and 999 sharing the same constellation), similar interpretations could be made with no need of these confusing maps.

Response

According to the comments we removed Figures 10 from the main text of the Manuscript to not overlap the text. We used the comment to correct the text in Results section. However we would like to ask Reviewer and Editor to keep the Figures in Supplementary as an additional visualization, as some readers could address to the Figures. We moved them to the Supplementary.

 Lines 406-408. Why do the authors state that the given amino acids in the polymerase segments suggest the low replication of the virus? Further, the following lines (408-414) do not provide a satisfactory explanation. And it is obvious that these H12 viruses are Low Pathogenicity Avian Influenza viruses, not only from the experimental infection of chickens, but also from the fact that a multibasic HA cleavage site has never been observed in viruses other than H5 or H7.

Response

We agree with a comment. We toned down the statement that observed substitution could just suggest the difference of replication activity of the virus in different models. However it is important that total combination of amino acid substitutions and main marker – cleavage site gave the LPAI phenotype of all viruses. Pathogenicity is polygenic characteristic and some novel mutation theoretically can give increasing of virulence. we confirmed LPAI phenotype by animal models.

We corrected the text – please see.

 The discussion section is much too long, unnecessarily complicated, and contains several redundant paragraphs (for exemple lines 439-40 are repeated in lines 514-19). It would be sufficient, and more informative, to discuss the main points (above in this review) in a synthetic manner.

Response

Completely agree with Reviewer. We shorten the Discussion section, tried to generalize one. We removed large part about avian groupings and migration changing for several words about main point in synthetic manner. We incorporated  useful comments from Reviewer – see above.

Totally we significantly reduced Discussion from previous 2093 words to 1385 with kind help of Reviewer. I would like to thank Reviewer for very strong and accurate work. I believe it will help the readers to get valuable information.

Lines 439-442.   “… is an inter-continental reassortant..”. Further, “active” and “actively” are excessive in the statement  “this rare subtype actively circulates...and is actively involved…”. Further again, it is not that particular subtype (i.e. the H12N2 subtype), but rather the H12 genomic segment.

Response

We removed the excessive statement: words  “active” and “actively”.

We corrected “inter-continental reassortant” and “H12”  instead of “H12N2” according to the comment

 Lines 453-467.  Fig S1. The alignment is somewhat confusing. Firstly, the colors here are useless. Secondly, the sequences should be grouped by similarity (i.e A/43 and A/1017 grouped; 1377 and 324 grouped). Thirdly, it would be more readable with one consensus sequence, and the six sequences aligned to the consensus (rather than choosing A/1377 as the reference sequence). 

Response

We removed useless colors and grouped the sequences according to the comment. The figure shows the substitutions between our strains to replace the section of the corresponding table S1, in here we believe it is appropriate to align the strains between themselves, not taking the consensus. Alignment just fully reflects the presence of a large number of substitutions (for this reason we do not give them in the table) between the two groups of sequences (A/1377+A/324+A/962+A/999) and (A/43+A/2017), which corresponds to the results of phylogenetic analysis.

 Minor remarks

Column 3 in table 2 is unnecessary (the values correspond to [column4 – column2]).

Response

We removed the Column 3 according to the comment

 Line 468. …long-distance migratory birds..(instead of distant migrants)

Response

Corrected according to the comment

Reviewer 2 Report

The authors answered all the questions. This review has further comments.

Author Response

I would like to thank Reviewer for very strong and accurate work. I believe it will help the readers to get valuable information.

Round 2

Reviewer 1 Report

The authors have adequately modified their manuscript according to the remarks that were raised. However, there remain a few minor remarks, as detailed below.

Line 55. …(HPAIVs) mostly originate from…

Line 58. …after genetically engineered introduction…

Line 63 …and infect mammals

Lines 418-419… belong to the North American lineage and the other five belong to the Eurasian lineage…

Line 427 …our five Eurasian H12N5 viruses…

Line 435… the NAs of the five H12N5 viruses…

Line 449… Our present data are consistent…

Lines 481-483… according to the phylogenetic analysis, ….related to segments that belonged to some HPAI strains

Lines 506-509. Ambiguous. Do the authors speak of the H12 viruses, or of the H12N2 isolate? 

Sentences that are either incomplete or grammatically incorrect: lines 178-79., lines 358-66

Author Response

Dear Reviewer,

Thank you once again for your useful comments and questions which helped to improve the manuscript.

Please see below each point for a detailed response on how we incorporated your feedback into the manuscript.

Line 55. …(HPAIVs) mostly originate from…

Response

Corrected according to the comment

Line 58. …after genetically engineered introduction…

Response

Corrected according to the comment

Line 63 …and infect mammals…

Response

Corrected according to the comment

Lines 418-419… belong to the North American lineage and the other five belong to the Eurasian lineage…

Response

Corrected according to the comment

Line 427 …our five Eurasian H12N5 viruses…

Response

Corrected according to the comment

Line 435… the NAs of the five H12N5 viruses…

Response

Corrected according to the comment

Line 449… Our present data are consistent…

Response Corrected according to the comment

Lines 481-483… according to the phylogenetic analysis, ….related to segments that belonged to some HPAI strains

Response Corrected according to the comment

Lines 506-509. Ambiguous. Do the authors speak of the H12 viruses, or of the H12N2 isolate?

Response We meant the H12N2 virus strain. So we corrected the sentence.

Sentences that are either incomplete or grammatically incorrect: lines 178-79., lines 358-66

Response

Line 178 - That was mistake – it is one sentence that contains the explanation what Bayes factor of 3 or more means

Lines 358-66: We tried to modify grammatically incorrect part and changed the text.

This manuscript is a resubmission of an earlier submission. The following is a list of the peer review reports and author responses from that submission.

Round 1

Reviewer 1 Report

In this manuscript by Kirill Sharshov et al., the authors genetically characterized six H12Nx avian influenza viruses that were isolated over a ten year period in northern Eurasia. This led the authors to propose a phylogeographical scheme for reassortment events, taking account of the migration flyways. 

The main findings are:

-  Five H12N5 and one H12N2 avian influenza viruses were isolated from wild birds of the Anatidae family in 2017 and 2018 in Northern Eurasia. Their full genomes were sequenced.

They all showed a Low Pathogenicity phenotytpe, as shown by the Intra Venous Pathogenicity Index (IVPI=0). They also showed no pathogenicity for mice (after intra-nasal inoculation with 10^6 TCID50).  They were all sensitive to the neuraminidase inhibitor oseltamivir.-          The H12 hemaglutinin of five of the six viruses were antigenically similar, as indicated by the cross-reactions in Hemaglutination Inhibition tests. However, A/1377 (i.e. the H12N2 virus from Russia far east) showed a fourfold reduction in HI titer, revealing an antigenic distance.-          The genomic constellations of the five H12N5 viruses were very similar, with their eight segments belonging to the Eurasian lineage. However, two among these five H12N5 (A/1017 and A/43) distinctively harboured a NS segment of allele B (Figure 1). The genomic constellation of A/1377 (the H12N2 virus) was clearly distinct, with four segments (H12, N2, NP and NS) belonging to the North American lineage. The HA sequences have a monobasic cleavage site, as expected, and they show a specificity towards avian receptor. 

The manuscript contains some interesting information, but at this stage it is preliminary and lacks important information regarding the geographical movements of the genomic segments. There are a number of points that need to be addressed by the authors, the major point being the phylogeographic analysis that is not at all convincing. 

Major remarks

Lines 353-54. These two lines contradict what is shown in Figure 1, where N2 of the H12N2 virus belongs to the North American lineage.  What exactly do the geographical maps represent? (Fig 10a to 10e and lines 351-363). The figures are entitled “Disseminations of the 8 segments….”. I understand that this means that each segment can subsequently be found in a posterior isolate. However this is probably impossible at present because the viruses studied here were isolated recently in 2017 and 2018, and likely there is a low probability of finding posterior isolates (in 2018 and 2019) that harbour segments originating from the studied viruses. Perhaps it is possible to elucidate how the genomic constellations of the studied viruses were arrived at, i.e. for each segment to find the most closely related segment in viruses isolated before the studied viruses. Perhaps this is what the authors tried to do, but it is not clear at all.  Line 388. What exactly is the meaning of the S42 residue? S42 is absolutely conserved in NS1 of allele A (with only very rare exceptions), while allele B-NS1 harbors A42. Therefore S42 in allele A and A42 in allele B is exactly what is expected, it is not a virulence marker. P42S is not associated with increased virulence (Table 4); instead S42P leads to a dramatically reduced virulence. 

The discussion section is an accumulation of assumptions regarding each and each virus, with no generalizable meaning. The two sentences at lines 550-51 are not at all clear and are not convincingly supported by the data. 

Minor remarksLines 102 and 105. replace 106.0 by 10^6.0. Similar remark for line 193.Line 109. replace 106 by 10^6

Reviewer 2 Report

In present study, Sharshov and colleagues isolated 5 H12N5 and 1 H12N2 avian influenza viruses from wild birds in three different geographic regions of Northern Eurasia area. They showed that all six H12Nx viruses are low pathogenic avian influenza viruses. Phylogenetic analysis showed that all five H12N5 contain Eurasian lineage segments. However, the H12N2 virus was a double reassortant virus containing HA, NS, and NP from American lineage and other segments are from Eurasian lineage. This manuscript supplies some value information about H12Nx avian influenza virus which are rarely isolated.

Major comments to the authors.

The organization of the Discussion section is confusing and long. Please concise and state with a more logical way. For the figure 10a-e, except the three locations where the H12Nx isolated, other dominant area or locations also should be labelled for better understanding the circulation of different gene segments between different flyways. The authors conclude that the H12Nx virus was rarely isolated during the 10-year surveillance. But during the 10 years surveillance, about 5000 ducks were tested, that is only about 500 samples per year. And the authors claim that a very wide territory from the Caspian Sea to the Pacific Ocean was tested, this indicates that only a very few samples were taken in one spot. So, the rare isolation of H12Nx may be not due to the rare of the virus, but not enough samples were tested. In P2 line 65, the authors claimed that the viruses were isolated from surveillance during 2016-2018. No H12Nx was isolated from other years? 60 viruses were isolated from 1500 samples during 2016-2018 (again indicating about 500 samples were tested a year) and 10% were H12Nx, I do not think the ratio is low. What are the other viruses and how about the isolation rate? This is important because the results may cause the change the conclusion.

Minor comments to the authors.

P1 line 26, the H12N2 should be double reassortant based on the information introduced. A/shoveler/Novosibirsk region/999k/2018(H12/N5) and A/mallard/Novosibirsk region/999k/2018@2018-10-01 were used across the manuscript including tables and figures. Please consistent the names. P4 line 148, what is the GISAID numbers for A/999 and A/962? P5 line 193, typos (107.8–3 EID50/mL) and MDCK cells (105.0–105.8 50% TCID50/mL), 7.8, 8.3, 5.0, and 5.8 should be superscript. P5 line 208, no HRI used in Table 2. P11, line 284, allele B stains not only belong to the Eurasian genetic lineage, also contain north American lineage based on figure 5. P15 line 344-346, the statement of “MP sequences of A/962-Eur and A/999-Eur from one side and A/teal/Russia_Primorje/18-1377/2018 from the other side belong to related phylogenetic subgroups” is confusing. What do you mean one side and the other side? P17 line 377, FLU-1377 should be consistent with A/1377. P17 line 389, based on the Table 4, PB1-F2 of A/962 and A/999 should be 66S.